# The Treatment Experience of Anorexia Nervosa in Adolescents from Healthcare Professionals’ Perspective: A Qualitative Study

**DOI:** 10.3390/ijerph20010794

**Published:** 2023-01-01

**Authors:** Yu-Shan Chang, Fang-Tzu Liao, Li-Chi Huang, Shu-Ling Chen

**Affiliations:** 1Department of Nursing, China Medical University Hospital, Taichung 406040, Taiwan; 2Department of Public Health, China Medical University, Taichung 406040, Taiwan; 3Department of Nursing, Hungkuang University, Taichung 433304, Taiwan; 4School of Nursing, China Medical University, Taichung 406040, Taiwan; 5Department of Nursing, China Medical University Children Hospital, Taichung 404333, Taiwan

**Keywords:** anorexia nervosa, adolescents, healthcare professionals, treatment experience, qualitative research

## Abstract

Anorexia nervosa (AN) is a serious psychiatric illness with a high mortality rate and a poor outcome. AN in adolescents can be difficult to treat. The prognosis of patients with AN depends highly on how early specialized AN treatment starts. Therefore, the purpose of this study was to explore the experiences of healthcare professionals in treating adolescents with AN. A qualitative study approach was conducted. Through semi-structured interviews, 16 healthcare professionals, including 10 nurses, 3 dieticians and 3 physicians from the paediatric ward at a university-affiliated medical centre in central Taiwan, shared their experiences. Recorded interviews were transcribed and analyzed by content analysis. Five themes and ten subthemes were identified: 1. Building a trusting relationship first: (a) spending time to build trust with the client and (b) establishing a relationship with the client’s parents; 2. The key to treatment success: (a) Clients’ awareness of the illness and (b) parents’ support for clients; 3. Consistency of team treatment goals: (a) maintaining stable vital signs and (b) achieving caloric intake; 4. Empowerment with knowledge about anorexia: (a) continuing education for healthcare professionals and (b) interdisciplinary collaborative care; and 5. Using different interaction strategies: (a) the hard approach and (b) the soft approach. In conclusion, the findings will provide important information for healthcare professionals to apply in monitoring the psychological and emotional states of adolescents with AN. The findings indicate that healthcare professionals should invite parents to participate in the treatment, support and guide them in their adolescent care, develop scales of family stress and support for AN in adolescents, develop interventions, and establish an early therapeutic alliance.

## 1. Introduction

Anorexia nervosa (AN) is a serious illness characterized by self-induced underweight, body image distortion and fear of weight gain [1,2,3,4]. AN is one of the most common chronic diseases in teenage girls, ranking third after obesity and asthma, and continues to rise in that population [5]. Its treatment is long and complex, involving a multidisciplinary team [6,7]. AN begins most often during adolescence and affects mainly adolescent girls and young women [8,9]. With scientific research, the disease of anorexia has gradually been understood, and there are many clinical records available [10,11,12,13]. It is also understood from a psychiatric perspective [8,14] and a body perspective (somatic perspective) [15]. Scientists have also identified biological correlates, multiple risk factors and patterns of aetiology, particularly sociocultural, developmental and pathophysiological [16,17], and genetic [18,19] types. Although the understanding of the causes of anorexia has increased, the psychopharmacological treatment of patients with anorexia is still ineffective [20]. The patients are difficult to treat, and the prognosis and outcomes are often disappointing. The problems include patient ambivalence towards change; resistance to treatment; comorbidities and personality traits; and cognitive impairment [21]. Studies have found that medical providers are dominated by biomedical theories and emphasize the visible signs of anorexia, namely, the patient’s physical condition, weight and behavior [3,22,23]. Psychological well-being appears to be a central criterion for eating disorder recovery, in addition to the remission of eating problems [24]. However, even though adolescents focus on their mental states and emotions, most professionals do not address their emotional issues and the adolescents’ wish to be seen as a whole person [22,23]. The patients often feel that the health care focuses too much on physical recovery and on the normalization of eating and weight. This approach can be perceived as unempathetic and give patients the impression that the therapists do not understand the patient’s real problems [23]. Trusting relationships with healthcare professionals are also considered important in developing the motivation to seek and stay in treatment [25]. Despite the well-known practical challenges in providing intensive treatments to individuals with AN, qualitative research into the perspectives of healthcare professionals and their treatment experiences is limited. Therefore, this study used qualitative research to explore the experiences of healthcare professionals in treating patients with anorexia, including understanding healthcare professionals’ therapeutic relationships with AN patients and parental involvement in the treatment process. These experiences can be used as a reference for healthcare professionals to care for patients with anorexia, and improve anorexia-related care.

## 2. Methods

### 2.1. Design

A qualitative study approach was used to describe and understand the in-depth complex phenomena associated with healthcare professionals, including nurses, dieticians, and physicians. Purposive sampling was used to recruit healthcare professionals who met the following criteria: (1) experience in caring for adolescents with anorexia nervosa; and (2) agreement to participate in this research.

### 2.2. Setting and Patients

This study was conducted in a general paediatric ward at a children’s hospital. Adolescents with AN who have life-threatening conditions are admitted to our hospital for medical stabilization. The length of hospital stay is about three to four weeks.

### 2.3. Participants

Healthcare professionals were invited to participate in this study through invitation letters. They were asked to complete a paper-based survey to indicate their willingness to participate in an interview. The inclusion criteria for healthcare professionals to participate in this study were at least one year of work experience in the hospital, and experience in taking care of at least one hospitalized adolescent with AN. The study included 16 healthcare professionals, including 10 nurses, 3 dieticians, and 3 paediatric gastroenterologists. The nurses were all female with a mean age of 30.2 years, ranging from 26 to 40 years; 3–11 years of clinical experience; and experience with caring for 1–8 adolescent patients with AN. The three dietitians were two females and one male with a mean age of 39 years, ranging from 38 to 41 years; 10–13 years of clinical experience; and experience with caring for 2–10 adolescent patients with AN. The three paediatric gastroenterologists were males with a mean age of 50 years, ranging from 33 to 59 years; 8–34 years of clinical experience; and experience with caring for 10–27 patients with AN.

### 2.4. Data Collection

Data collection consisted of semi-structured interviews, based on the following questions: (1) Can you tell me about your experience in the treatment and care of adolescents with AN? (2) Can you tell me how to develop a therapeutic relationship with an adolescent with AN? What was your approach? (3) Can you tell me how to guide parents in caring for their children? How can one support and incorporate parents’ ideas? (4) Can you tell me how the medical team can be trained or educated to improve the care of AN? Data collection was conducted from January to June 2019. Each interview was recorded, coded, and analyzed verbatim, and the experience was shared and discussed with experts trained in qualitative research. Data collection was completed using the data saturation criterion through the repetition of information in the statements from a total of 16 in-depth interviews. The interviews ranged in length from 45 to 60 min.

### 2.5. Data Analysis

Data were analyzed with the qualitative content analysis approach from Graneheim and Lundman [26] by determining meaning units, condensed meaning units, codes, subcategories, categories, and themes. The interviews were carefully transcribed, and each transcription was read several times to obtain a general understanding of the interview. Then each text was read line-by-line and meaning units were identified, after which each meaning unit was condensed and assigned a code. Next, codes were assigned to subcategories according to their similarities and differences. Similar subcategories together formed categories, and finally, themes emerged.

### 2.6. Trustworthiness

To ensure the trustworthiness of the study, our procedure followed the guidelines proposed by Lincoln and Guba [27]. The credibility of the data was enhanced by the authors’ expertise in nursing and qualitative research, which allowed us to fully understand the healthcare professionals’ experiences regarding the treatment and care of adolescents with AN. Transferability was facilitated through the use of purposive sampling, and the data sources were enriched by including participants of different professions, ages and care experiences. Dependability was promoted by the authors meeting frequently to discuss the data analysis and by checking and rechecking the labelling, sorting and naming of themes during data analysis for verification, as suggested. Confirmability was ensured by describing the entire research process and procedures in detail, keeping a reflexive journal, and maintaining an audit trail [27]. Healthcare professionals’ experiences were extracted as thick descriptions of the related phenomena.

### 2.7. Ethical Considerations

This study was performed in accordance with the principles of the Declaration of Helsinki. All the study procedures were approved by the Hospital Human Investigation Committee at each of the medical centres (IRB No. CMUH106-REC1-132(AR-1)). The participants’ rights of anonymity, confidentiality and withdrawal from the study were explained at the time of the interview.

## 3. Results

### 3.1. Participants’ Characteristics and Identified Themes

The study included 16 healthcare professionals, including 10 nurses (all female, aged from 26 to 40 years), 3 dieticians (all female, aged from 38 to 41 years), and 3 physicians (all male, aged from 33 to 59 years), for a total of 16 participants. 

The findings were categorized into five major themes and ten sub-themes: 1. Building a trusting relationship first, including the sub-themes: (a) spending time to build trust and (b) establishing a relationship with the client’s parents; 2. The key to treatment success, including the sub-themes: (a) awareness of the illness and (b) parents’ support; 3. Consistency of team treatment goals, including the sub-themes: (a) maintaining stable vital signs and (b) achieving caloric intake; 4. Empowerment with knowledge about anorexia, including the sub-themes: (a) continuing education and (b) interdisciplinary collaborative care; and 5. Using different interaction strategies: (a) the hard approach and (b) the soft approach. The main themes and sub-themes are presented in Table 1. 

### 3.2. Theme 1. Building a Trusting Relationship First

The need to build a trusting relationship is based on the fact that healthcare professionals (including nurses, dieticians, and paediatric gastroenterologists, referred to as physicians) cannot easily establish a therapeutic relationship with a client at the first encounter. In the beginning, the client is highly defensive and reluctant to express her thoughts and feelings. She engages in little social interaction and exhibits apathy and rigid behaviors. Thus, the establishment of the initial trusting relationship is very important. Once such a relationship is built, follow-up treatment can ensue. This theme included two subthemes: (a) spending time to build trust with the client and (b) establishing a relationship with the client’s parents. The first subtheme, “spending time to build trust,” shows that the establishment of trust requires spending time to gain her trust, understand her, make her dependent on us, and slowly persuade her to share her innermost thoughts. Three physicians identified the importance of a trusting relationship.

As one physician said, “*You must take the time to establish a relationship with her. She is willing to rely on you, and she is willing to tell you where the problem is. Slowly change her mind and see if she can recognize that food refusal is not good for her.*” (C2).

Physicians also believe that the characteristics of nurses are very important, and that they are willing to spend time talking to and understanding the client. As one physician said, “*The characteristics of nurses are very important. They may not be able to take care of them if they don’t have the right characteristics. You need to screen nurses. Like some discharged cases, nurses are also willing to contact them. Because my (the physician’s) time with her is limited, the nurse has more time with her, especially some night nurses*.” (C1). However, for most nurses, the routine care to be performed every day is multifarious and complicated, and the work schedule is tight, with an average of 6–7 patients per day. It takes a lot of time to perform general care. When facing patients with anorexia, they need sufficient time to understand the mental illness. As one nurse said, "*It often takes more time to establish a relationship with patients with anorexia, but the usual nursing work is already quite busy. So there is no way to deeply understand the case, and it is more difficult to build therapeutic relationships*.” (C13).

The second subtheme, ‘establishing a relationship with the client’s parents’, refers to the fact that, since it is difficult to establish a relationship with the client at the beginning, healthcare professionals can first establish a relationship with the client’s parents to understand the client’s preferences. As one nurse said, “*Initially, anorexia patients are alienated from us; they don’t even want to talk to us. So I choose to establish a relationship with the parents of the case first. And then I learn about the reasons for the client’s anorexia, her thoughts, and her preferences from the parents. Knowing that the client likes the punch-in food on Instagram (IG), the Lala Bear, etc., I will go up to see what food is available near the hospital, and then recommend it to her during treatment*” (C13).

It is also important to let parents know that family relationships are very important. As one participant said, “*We still emphasize that parents should give children enough love and tolerance, and they will be easier to deal with and less likely to come to this point.*” (C2). Some doctors also hope that parents will realize that it is not appropriate to blame the child. They must understand that the child is ill, and her thoughts will be affected by the disease. For example, one participant said, “*In fact, the client is in a state of illness, her thoughts will be affected physiologically. If the parents are willing to accept it, it will be better, instead of blaming the child all the time.*” The participant also stated, “*We can imagine that their parents and families must be in conflict with anorexia, because the parents must have hurriedly forced her to eat and dragged her to see the doctor*.” (C1).

### 3.3. Theme 2. The Key to Treatment Success

This theme includes two sub-themes: (a) Clients’ awareness of the illness and (b) parents’ support for clients. The first theme, “Clients’ awareness of the illness,” means that the most difficult aspect of the treatment and care of adolescents with anorexia is whether the patient understands that they have an illness. A patient who is aware of the illness understands that the disease causes physical problems and can generally gain weight through treatment. In contrast, adolescents who do not perceive an illness do not think they have a problem, but their parents take them to a doctor for treatment. In general cases of this lack of awareness of the illness, when the physician asks, “*Why did your parents take you to see a doctor? Why did you come to see a doctor?*”, the patients will often answer, “*I’m fine, it’s my parents who asked me to see a doctor*.” (C1). One physician also said, “*In fact, these children don’t have enough awareness of the illness. I told (one girl) to go to the intensive care unit, but she would not agree … At this time, her heartbeat had dropped to 28, and she didn’t think anything was wrong with her.*” (C3).

Another dietician said, “*We compared children who lacked awareness of the disease, and they would say that they should not eat this and should not take this IV, because there are too many calories*.” (C14).

The second subtheme, “parents’ support for clients”, is very important. Three physicians believe that parents should give their children enough love and tolerance, and they should not blame the child, because the child is currently in a state of illness. Therefore, her thoughts will be affected by the disease. One physician mentioned that “*the most difficult part of treating anorexic teenagers is not only that the patient has no awareness of the illness, but also, if the parents’ support is not enough, sometimes it is very difficult to treat anorexia.*” (C3). It is also very important to let parents know that family relationships have great significance. One physician said, “*We still emphasize that if parents give children enough love and tolerance, they will be easier to deal with and less likely to come to the end of the disease. Family relationships and support are still very important.*” (C2). Three physicians also hope that parents will not blame the child in an attempt to change the child’s behavior. One physician said, “*The child is sick now, not because she is unwilling to cooperate or skip meals, but because her mind is affected physically. If the parents are willing to accept it, they will not be blamed all the time*.” (C1).

### 3.4. Theme 3. Consistency of Team Treatment Goals

This theme included two sub-themes: (a) maintaining stable vital signs and (b) achieving caloric intake. The first subtheme, “maintaining stable vital signs,” refers to the fact that healthcare professionals’ treatment goals must start with physiology, as it is believed that nutritional deficiencies can affect cognitive function. First, the client’s vital signs should be stable. As one physician said, “Anorexia nervosa is most commonly [associated with] a very low heartbeat (30 bpm), or low body temperature (35 °C), low blood pressure, electrolyte imbalances. These are life-threatening conditions, and sometimes they have to be admitted to the ICU, which may be mandatory.” (C1). The second subtheme, ‘achieving caloric intake’, is about focusing on the patient’s dietary intake and weight growth. One dietician said, “We will start by increasing the amount of food she eats at each meal, and try to add to it slowly, and maybe reach her caloric intake. We may use other intravenous methods, or even use nutritional supplements to intervene, and to pull her whole weight up!” (C14). The nurse and the client set a weight goal together, and upon achieving the goal, the client will be allowed outside or discharged from the hospital. One nurse said, “Give her a goal, how much I want you to eat every day, and if you gain this much weight, I can give you time off, and your mother can take you out for fun. And if you gain that much weight, we will let you out of the hospital.” (C4). 

### 3.5. Theme 4. Empowerment with Knowledge about Anorexia

This theme refers to the ways medical team members can improve collaborative care and their related knowledge of anorexia, and then continuously improve their care. This theme includes two sub-themes: (a) continuing education for healthcare professionals and (b) interdisciplinary collaborative care. The first theme, “continuing education for healthcare professionals,” describes the lack of knowledge of healthcare professionals, especially nurses and dieticians, in the care of anorexia and the expectation of continuing education related to anorexia. One physician said, “*Our care for anorexia is taught by the attending physician one by one, from the intensive care unit to the ward care, and then to the outpatient care. In fact, education is carried out during the follow-up process and the ward rounds. This kind of education only means that the few people who are cared for know how to take care of them. Nurses still don’t know how to care of them.*” (C3). One dietician reflected on the need to improve her knowledge of psychology. As she said, “*We should study psychology. Anorexia is not only physical; the psychological part also plays a big role in the treatment.*” (C16). The second subtheme, ‘interdisciplinary collaborative care’, reflects the fact that most nurses feel that they lack experience in caring for patients with anorexia, and they look forward to continuing education in this area or sharing their caring experiences in the ward. For example, one nurse said, “*We can invite psychiatrists to do ward teaching activities to teach us how to face and how to care for people with anorexia, what conversations and behaviors are helpful to them, and what to avoid*.” (C13) Another nurse said, “*Interdisciplinary discussions can be held, and paediatricians, psychiatrists, social workers, dieticians, and nurses can be invited to discussions, and individual discussions can be conducted on individual cases to provide patient-centred care.*” (C10). Physicians expect a dietician to function not only in nutritional assessments but also in psychological assessments. As he said, “*If a dietician falls into a machine and it looks like she’s taking care of a machine, she doesn’t take into account the whole body and mind of the patient. What the patient needs, such as providing what to eat, should not only focus on nutrition, but also on whether it tastes good or not!*” (C3). 

### 3.6. Theme 5. Using Different Interaction Strategies

This theme includes two sub-themes: (a) the hard approach and (b) the soft approach. The first theme, “the hard approach,” refers to the fact that some participants would use coercive methods to require patients to increase their caloric intake and achieve body weight goals. Most children fear the insertion of the nasogastric tube the most, so one physician said, “If you stop eating again, I will put you on a nasogastric (NG) tube. I will force-feed you, I will transfer you to the intensive care unit, and then put the TPN, and I will force you to be given nutritional injections.” (C3). Even placing the NG tube where the child can see it can be effective; as the nurse stated, “put the nasogastric tube at the end of the bed.” (C5).

The second subtheme, ‘the soft approach’, describes the approach of some participants of using gentleness or physical comfort, combining euphemisms with a strict manner, creating an agreement with the patient, and expressing empathy. For gentleness or physical comfort, one physician said, “*I use chatting, and every time I go in, she will chat for more than 30 min. And when I go in, I will hug the child and touch her hand. Then I’ll tell her that she can tell me anything, and also allow her parents to be by her side.*” (C2). From the physician’s point of view, a massage strategy can be useful for building a trusting relationship. One physician stated, “*I also do a lot of things that general doctors can’t do. I sit down and talk to the children…I will give the children massage, and they have no flesh but bones, then the parents will be by their side, and gradually I will teach the parents. I’ll say, when you are in contact with the child, give her a massage and physical touch when you have time, because if you don’t develop a little relationship with her, she will never make progress.*” (C2) Regarding the use of euphemisms with a strict manner, as one physician said, “*I am more euphemistic. I don’t scold people, but at this time, it is necessary to make comparisons. If she does not cooperate, I may need to be stricter and tell her clearly that she can’t do this, and if she doesn’t cooperate, the less I will bargain with her.*” (C3). Regarding creating an agreement with patient, as one nurse said, “*The doctor made us draw up three rules with the child: not to self-harm, not to hurt the parents, and not to engage in violent behavior; otherwise, she would be transferred to the psychiatric ward.*” (C5). Regarding empathy, as one nurse said, “*At the beginning of taking care of such a case, when I spoke, she would just stare at me and her hostility was very strong. I would empathise in the here and now; if I were her, what would I want?*” (C4).

## 4. Discussion

The main objective of the present study was to explore the experiences of healthcare professionals in treating adolescents with AN by interviewing physicians, nurses, and dieticians. Our results showed that, beginning at the patient’s admission, it was crucial to build a trusting relationship in the treatment process by spending time to understand the patient and establish a relationship with their parents. This finding is similar to those of previous reports [28,29], which reported that nurses perceived trust as important in the care of people with AN, as it was essential for guiding changes in the patients’ behaviors. Trust was described as a component of the relationship that could be developed over time [30,31]. It was also consistent with Gulliksen et al. [30], who emphasized the importance of the therapeutic “relationship” in the treatment of chronic AN. That is, one therapeutic aspect of patient admission is establishing a ‘secure base’ in individuals where one is lacking. As Ross et al. [32] found, patients reported that positive encounters were ones that made them feel understood, safe, and valuable, while also putting them in a better mood. The phrase “like home” was used to express positive encounters with health professionals. Therefore, establishment of the therapeutic alliance is a fundamental activity in mental health nursing [33], and an important aspect of care when working with people who have AN, with implications for the outcomes [34,35].

Anorexia nervosa is a psychiatric disorder with a considerable risk of serious physical morbidity, and even death. Healthcare professionals must look at extremely thin, malnourished bodies [36], so the team treatment goal of healthcare professionals in this study consistently focused on (a) maintaining stable vital signs and (b) achieving caloric intake. This finding is similar to a previous report [37] that, when it is important to stick to a task (e.g., weight gain in anorexia nervosa), the clinician should be consistent in pushing the need to achieve that goal. As Sibeoni et al. [22] found, healthcare professionals can then have great difficulty focusing their attention on what the adolescent thinks and feels about it. This issue is undoubtedly more serious within inpatient units. Therefore, healthcare professionals tend to focus on the visible signs of anorexia nervosa, namely, the condition of the body and the patient’s weight, while adolescents focus on their psychological state and their emotions. The clinician and the patient focus on different points, which can easily lead to conflict. Bourion-Bedes et al. [35] suggest that, to establish an early therapeutic alliance, healthcare professionals need to pay attention to the psychological and emotional states of these adolescents and try to approach how the adolescents live the disease from the inside.

The healthcare professionals in this study perceived that the key to treatment success was the patient’s awareness of the illness and the parents’ support. Although it is difficult for a patient with AN to recognize her own disease, parental support is so important that treatment of anorexia nervosa in adolescence should always involve the parents [38]. They too must initially focus on issues important to the teen; that is, the individual psychological aspects of the disease. In fact, this is the most important aspect for the adolescent’s involvement in his or her treatment, and parents have an important role to play in this step of the treatment process.

Regarding the component of knowledge about anorexia, the nurses and dieticians in this study perceived that they lacked knowledge about caring for patients with anorexia. Therefore, they had an expectation of continuing education related to anorexia. This finding is similar to previous reports [39,40] that there is a need for adequate education, training, support and preparation for dealing with patients with AN with greater understanding. As Wu & Chen [41] found, nursing staff generally lack positive feelings about patient care for AN and even question their abilities to provide quality nursing care. Ramjan [40] also reported that, when nurses lacked knowledge about the illness, this mindset facilitated an overly narrowed focus on the patient’s behaviors, resulting in a power struggle in the relationship. Therefore, Ramjan [40] suggested that education for these ‘specialist’ nurses needs to include an understanding of the elements, stages and turning points in recovery from anorexia nervosa, and not only the understanding of its symptoms and aetiology.

Regarding the use of different interaction strategies, our study found that healthcare professionals will interact with patients with AN in different ways to build a trusting relationship and increase their caloric intake. The hard approach was mentioned in a previous study [41], indicating that, when a doctor discusses a patient’s condition in a threatening way, the patient will fear the consequences of punishment and passively accept treatment. As a result, patients cannot entirely trust the healthcare professionals, and the healthcare professionals and patients are suspicious of each other. Similarly, Geller & Srikameswaran [42] indicated that, although clinicians may at times have to implement interventions which are not wanted by the patient, having a predetermined contract of non-negotiables, reiterating the rationale for these non-negotiables, and attempting to offer as much choice as possible can help to maintain trust and the therapeutic alliance.

Moreover, our study also found that participants described interaction strategies with a soft approach (gentleness or physical comfort, combining euphemism with a strict manner, creating an agreement with the patient, and empathy). These findings are similar to those of Geller & Srikameswaran [42], who indicated that nursing approaches may vary between the unwavering strict and stern approach to the democratic, negotiation-based and non-pressurised approach. In addition, our study also found that a massage strategy could be used for building a trusting relationship. This finding has not been mentioned in previous studies. From this, it can be concluded that it is important to make the client feel as if she is cared for and regarded as an individual.

## 5. Study Limitations

First, this study recruited 16 experienced health professionals (3 paediatric gastroenterologists, 10 nurses, 3 dieticians) who had cared for patients with AN in central Taiwan. Therefore, the experiences presented here cannot be generalized to health professionals from other parts of Taiwan, or to health professionals in western countries. However, the experiences of the participants are powerful and add richness to the existing knowledge on health professionals’ experiences with patients with AN, and they provide motivation to rethink the design of in-service education for AN patient care. Second, the paediatric gastroenterologists were all male, while the nurses and dieticians were all female, and the dieticians were 2 females and one male. The sample of participants was very unequal, not only in gender but also in number (10, 3 and 3). It is recommended that the question of whether different genders have different care experiences be explored in the future. Finally, because patients under the age of 18 in Taiwan are admitted to paediatric wards, adolescents with AN will first be hospitalized in a paediatric ward for observation, and not in a psychiatric ward. In addition, paediatric nurses do not have experience in supporting each other across departments, so their care experiences may be limited.

## 6. Conclusions

Patients with anorexia have extremely fragile and malnourished bodies, and healthcare professionals can be shocked by the sight them—reduced to only skin and bones, these patients even face the risk of death. Healthcare professionals are therefore more focused on the visible signs of anorexia, namely, the physical condition and weight maintenance, and they find it more difficult to focus on the individual’s thoughts and feelings. All physicians will first spend time to build a trusting relationship with the patient, and they will address the patient’s emotional problems. If both are not simultaneously addressed, the risk of relapse and readmission after discharge is high. Thus, it is well established that both mental and physical recovery are very important in patients with AN. The results of this study should provide further understanding of the perspectives of medical professionals, ways to face the disease of anorexia, and ways to function as a team and to strengthen the team’s knowledge about anorexia.

## Figures and Tables

**Table 1 ijerph-20-00794-t001:** Themes and Sub-themes.

Theme	Sub-Theme
1. Building a trusting relationship first	1-1 Spending time to build trust with the client
	1-2 Establishing a relationship with the client’s parents
2. The key to treatment success	2-1 Clients’ awareness of illness 2-2 Parents’ support for clients
3. Consistency of team treatment goals	3-1 Maintaining stable vital signs
	3-2 Achieving caloric intake
4. Empowerment with knowledge about anorexia	4-1 Continuing education for health professionals
	4-2 Interdisciplinary collaborative care
5. Using different interaction strategies	5-1 Hard approach
	5-2 Soft approach

## Data Availability

The data are not publicly available due to restrictions regarding privacy and ethical considerations of the study participants.

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
