# Peer review of "The Treatment Experience of Anorexia Nervosa in Adolescents from Healthcare Professionals’ Perspective: A Qualitative Study"

_ijerph, 2023, doi:10.3390/ijerph20010794_

Round 1

Reviewer 1 Report

I would like to thank the Editor for the opportunity to review this study and I am flattered to be able to provide my contribution. In general, I find this article to be well written, I do find this paper to be a good discussion issue about the treatment experiences from healthcare professsionals in anorexia nervosa. However, the paper presents some weaknesses, and I would suggest reconsidering publishing this manuscript after minor revisions. Suggestions are reported in the following comments. I would ask the Authors to address minor amendments, as follows:

1. Introduction

The introduction is a bit brief; the authors could expand it a bit more.

2. Method -

The authors should include the healthcare professionals participating in this qualitative study, including years of professional experience, years of experience with AN, gender, age,...etc.

5.- Limitations

The authors should include that the sample of participants is very unequal, not only in gender as they point out, but also in number (10, 3 and 3).

I hope these comments are useful in moving your research forward.

Reviewer 2 Report

Intro:

The introduction is well written and provides enough information to lead to the aim of the study. The aim is formulated very general, maybe the research question could be phrased a bit more precise, especially regarding theme 2 (parents support)?

L33-35: this sentence seems to be a bit unprecise, what do you mean by that? Please clarify

Methods:

When reading the method section, following questions arised, please clarify:

Please state the language, the interviews were carried out

What was the exact procedure of the semi-structured interviews? (timeline: is the order of the questions in 2.2. data collection the order questions were asked?)

How exactly were the interviews transcribed and when was the transcription carried out?

I would like to suggest to add a category “participants” to the methods section:

-          How were participants recruited?

-          what medical disciplines was the group of physicians comprised of (general practicioners…)?

Results:

Table 1: I would suggest to restructure Table 1. It seems confusing wether or not sub-themes fit to themes.  Maybe you can also add the codes into this table?

Description of codes seem to be missing

L 120 – 125: this content seems to be an explanation for the themes. Who said that? Did the participants agree on that? If the purpose is to explain the theme, then I think it needs a reference here. If this statement is retrieved from the interviews, then please mark it as such.

L157: “some doctors”: please be more precise in stating how many of the interviewed persons had that opinion (throughout the whole results section, also L131)

L182: please state this in a more precise scientific language. Why is it very important and who thinks that it is very important?

3.4. Why is this section completely in italic?

L: 243 – 244 if it is a statement, please use italic

General comments on result section:

Please check again which phrases should be in italic and which should not be. There seems to be inconsistency.

 Discussion

Overall, the discussion is well written. When discussing study limitations, maybe you could also refer to the impact of sex (physicians male, nurses female) – how could have the sex difference impacted your results?

Conclusions

L 366 – 367 it is well established that both, mental and physical recovery is important in patients with AN. Please rephrase

Could you draw a conclusion for parents as well?
